# Enhanced Cell Wall and Cell Membrane Activity Promotes Heat Adaptation of *Enterococcus faecium*

**DOI:** 10.3390/ijms241411822

**Published:** 2023-07-23

**Authors:** Li Wang, Aike Li, Jun Fang, Yongwei Wang, Lixian Chen, Lin Qiao, Weiwei Wang

**Affiliations:** 1Academy of National Food and Strategic Reserves Administration, Beijing 100037, China; wl@ags.ac.cn (L.W.); lak@ags.ac.cn (A.L.); wyw@ags.ac.cn (Y.W.); clx@ags.ac.cn (L.C.); 2College of Bioscience and Biotechnology, Hunan Agricultural University, Changsha 410128, China

**Keywords:** *Enterococcus faecium*, heat adaptation, multi-omics, cell wall, cell membrane

## Abstract

*Enterococcus faecium* (*E. faecium*) is widely used in foods and is known as a probiotic to treat or prevent diarrhea in pets and livestock. However, the poor resistance of *E. faecium* to high temperature processing procedures limits its use. Strain domestication is a low-cost and effective method to obtain high-temperature-resistant strains. In this study, heat treatment was performed from 45 °C to 70 °C and the temperature was gradually increased by 5 °C every 3 days. After domestication, the survival rates of the high temperature adaptation strain RS047-wl under 65 °C water bath for 40 min was 11.5 times higher than WT RS047. Moreover, the saturated fatty acid (SFA) contents in cell membrane and the cell volume significantly increased in the RS047-wl. The combined transcriptomic, metabolomic, and proteomics analysis results showed a significant enhancement of cell wall and membrane synthesis ability in the RS047-wl. In conclusion, one of the main factors contributing to the improved high temperature resistance of RS047-wl was its enhanced ability to synthesize cell wall and membrane, which helped maintain normal cell morphology. Developing a high-temperature-resistant strain and understanding its mechanism enables it to adapt to high temperatures. This lays the groundwork for its future development and application.

## 1. Introduction

*Enterococci* are known as probiotics to treat or prevent diarrhea in pets and livestock [1]. This genus is the symbiotic lactic acid bacteria (LAB) in the gastrointestinal tract of animals [2]. *E. faecium* is widely used in foods for its relatively high thermal resistance [3] and its ability to grow in the presence of salt and low pH values [4,5] as well as a wide range of temperatures [6]. However, during the processing of spray drying and feed granulation, *E. faecium* faces a major challenge, which can result in cell death. Thermally robust strains are useful for the industry. The cell wall and cell membrane damage may be a possible reason [7]. Other possible reasons in the bacteria where damage may occur as a result of spray drying and heat stress include DNA [7] and ribosomes [8].

However, bacteria can adapt to environmental changes and get through various general and specific stress responses [9]. Heat stress is one of the most extensively studied topics for LAB [10,11]. Min et al. used a two-step adaptive laboratory evolution to obtain a heat-resistant strain, and the survival ratio increased by 20% [12]. The reported stress adaptation mechanism of LAB involves changes in membrane fatty acid composition [13], expressions of heat shock proteins [9], and activation of quorum sensing [14]. Such adjustments aimed to balance the effects of stress with the physiological needs of the cell so that critical physiological parameters were fine-tuned to guarantee survival.

In previous studies, we isolated a strain of *E. faecium* with an excellent bacteriostatic performance from poultry intestine, named RS047, and we conducted a modified continuous heat stress domesticating method to improve the heat resistance of RS047. The acclimated strain was named RS047-wl [15]. In this study, additional investigations were conducted to examine the physiological and biochemical characteristics of RS047-wl. Label-free quantitative proteomics, transcriptomics, metabolomics, and analysis of physiological indexes were performed to further explore the mechanisms underlying the strain’s resistance to high temperatures.

## 2. Results

### 2.1. Analysis of Physiological and Biochemical Characteristics of RS047-wl

#### 2.1.1. Evaluation of Heat Resistance of Strains for Industrial Production

Pilot scaled fermentation of 1-ton fermentation tank (*n* = 3) were conducted for RS047-wl, and the survival rate of the bacteria for the products increased from 47.95 ± 6.50 (%) to 90.29 ± 18.18 (%) under the heat stress for 10 min at 85 °C (Figure 1A).

#### 2.1.2. Growth Performance of RS047-wl

Growth performance of the WT RS047 and RS047-wl were studied and the growth curves were shown as Figure 1B. WT RS047 entered the logarithmic phase after 4 h and reached the plateau phase after 10 h, while RS047-wl entered the logarithmic phase after 5 h and reached the plateau phase after 11 h.

#### 2.1.3. Determination of Key Metabolic Enzyme Activities

The enzyme activity of lactic dehydrogenase was significantly declined in RS047-wl, and the activity of hexokinase, pyruvate kinase, Na^+^K^+^-ATPase, and Ca^++^Mg^++^ATPase had no significant change compared to WT-RS047 (Figure 2).

#### 2.1.4. Fatty Acid Composition of RS047-wl

The main fatty acid compositions for RS047-wl and WT RS047 are summarized in Figure 3. The unsaturated fatty acids (UFA) consist of palmitoleic acid (C16:1), oleic acid (C18:1), linolelaidic acid (C18:2), eicosenoic acid (C20:1), while the saturated fatty acids (SFA) include myristic acid (C14:0), hexadecanoic acid (C16:0), octadecanoic acid (C18:0), and arachidic acid (C20:0). This study found that the sum of the concentrations of SFA and the contents of C16:0 and C18:0 were significantly higher (*p* < 0.05) in RS047-wl compared to WT RS047. Meanwhile, the sum of the concentrations of UFA and the content of C18:1 were significantly lower in RS047-wl than those in WT RS047 (*p* < 0.05). Consequently, the SFA concentration increased from 44.14% (WT RS047) to 49.29% (RS047-wl). In contrast, the UFA concentration decreased from 55.85% (WT RS047) to 50.71% (RS047-wl).

#### 2.1.5. Morphological Observation of RS047-wl

The cell morphology of RS047-wl and WT RS047 was analyzed by transmission electron microscope (TEM) (Figure 4). The WT RS047 and RS047-wl cells exhibited a round shape and a smooth surface. Nevertheless, RS047-wl showed swelling under high temperature. Then, the membrane thickness, cell wall, cell length, width at undivided period, early division, and late division were measured (Figure 5). The cell widths of the RS047-wl were all significantly increased in three periods (*p* < 0.05) and the lengths of the cell were both increased in the early division and late division stages (*p* < 0.05), which led to the increased volume of cells. The cell membrane thickness of the mutant decreased significantly in the undifferentiated stage but returned to normal in the division stage. In addition, there were no significant changes in cell wall thickness.

### 2.2. The Differential Protein Abundances of WT RS047 and RS047-wl

To further investigate the regulatory mechanisms of heat resistance for RS047-wl, we isolated whole protein samples from WT RS047 and RS047-wl strains. Each sample was quantified using a label-free quantitative proteomics method to compare the differential expressed proteins between them. In this study, a total of 1633 proteins were identified by mass spectrometry. The condition for identification was set as having a unique peptide number ≥ 2 and a false discover rate (FDR) < 1%. Differentially expressed proteins were identified based on their abundance ratio of ≥1.2 (upregulated expression) or ≤0.83 (downregulated expression, along with a *p*-value < 0.05. A total of 43 differentially expressed proteins were identified, with 11 proteins being upregulated and 32 proteins being downregulated when compared to WT RS047. We generated a heat map of the differentially expressed proteins as shown in Figure 6. Furthermore, we used GO term enrichments to analyze altered proteins in both comparisons. In the analysis of biological process (BP) enrichment, the terms showing the most significant changes in RS047-wl were related to the defense response, immune system process, interspecies interaction between organisms, response to external biotic stimulus, defense response to another organism, and response to other organisms. Additionally, in the analysis of cellular component (CC) enrichment, the terms that showed the most changes were the extracellular region, the integral component of the membrane, and the intrinsic component of the membrane. In the molecular function (MF) enrichment, the terms that showed the most changes were amidase activity, protein-N(PI)-phosphohistidine-sugar phosphotransferase activity, and carbohydrate trans-membrane transporter activity. The protein LysM peptidoglycan-binding domain-containing protein participated in all the above BP, CC, and MF GO terms (Figure 7A).

Moreover, we utilized KEGG terms enrichment to analyze the proteins that were altered in both comparisons. The differentially expressed proteins were enriched in starch and sucrose metabolism. Additionally, the adhE protein was found to be associated with the categories of tyrosine metabolism, fatty acid degradation, and naphthalene degradation (Figure 7B). We found that the significantly upregulated proteins were Pepv, LysM, and YlaN. They all had functions related to cell wall synthesis and cell morphological change.

### 2.3. The Differential Genes Abundances between WT RS047 and RS047-wl

Whole genome mRNA sequencing was used to investigate the changes in gene expression between WT RS047 and RS047-wl. We searched for 12 genes related to cell wall and cell membrane metabolism that were significantly upexpressed in RS047-wl (Figure 8). Among them, *FabD*, *FabG*, and *gpsA* were related to cell membrane metabolism. *MraZ*, *mraW*, *murF*, *tagH*, *LtaS*, *prsA*, *LTA* synthase family protein, *prsA*, and *MltG* were related to cell wall metabolism.

### 2.4. The Differential Metabolite Abundances of WT RS047 and RS047-wl

In total, 1293 differential metabolites were identified. The expression levels of certain metabolites in the tricarboxylic acid cycle and purine metabolism were significantly decreased. In the field of proteomics and transcriptomics, the substances that exhibited the highest upregulation were primarily associated with cell wall and cell membrane metabolism. In metabolites, the significantly upregulated metabolites associated with cell membrane metabolism were glyceryl, 8-Amino-7-oxononanoate (KAPA), and phosphatidic acid (PA, 18:3(6Z,9Z,12Z)/20:1(11Z)).

## 3. Discussion

Feed granulation with high processing temperature is one of the most challenging hostile conditions for the probiotics in diet. Previously, we found that the survival rate of strain RS047-wl under 65 °C water bath for 40 min was 11.5 times higher than the wild strain [15]. In this study, the adapted strain RS047-wl showed 88% growth of survival rate under an 85 °C high temperature challenge.

The cellular membrane has been implicated as an important site in microbial thermal destruction, which not only functions as a physical barrier, but also plays a structural role in the cell. The proportion of SFA in the membrane significantly increased in RS047-wl. It has been reported that a higher proportion of SFA in the cell membrane resulting in reduced proton permeability [16]. The SFA would promote the interaction between acyl chains, decreased membrane fluidity, and increase the rigidity of cell membranes, thereby contributing to improved high-temperature resistance [17]. In addition, TEM observations demonstrated that cell membranes did not get thinner with significant enlarged volume of the cell, revealing an adaptive response to a stressful environment.

The cell membrane is mainly composed of phospholipids, proteins, and sugar. The phospholipid bilayer is the basic scaffold of the cell membrane. The *fabD* and *fabG* genes are the key enzymes for the synthesis of fatty acids (Figure 9). The *fabD* gene catalyzes malonyl-CoA to acyl-carrier-protein (ACP), and the *fabG* is a reductase catalyzing ACP in type II fatty acid. It has been reported that the over-expression of fatty acid synthesis genes were found to confer stress tolerance [18]. KAPA is the precursor for the synthesis of biotin, which is the precursor substance to synthesize fatty acids. Fatty acids and glycerol can be used to synthesize PA, which has a central role in lipid metabolism, and it is required for the de novo synthesis of all the glyceryl phosphatide and triacylglycerol. In addition, PA is known to facilitate signal events, membrane fission, and fusion events. The NAD^+^-dependent glycerol 3-phosphate dehydrogenase (Gpd, gene *gpsA*) is the rate-limiting enzyme of the glycerol biosynthetic pathway. The over-expression of genes *FabD*, *FabG*, gpsA, and the over-expression of glyceryl, PA, and KAPA enzyme genes, led to an increase in membrane synthesis. This guaranteed the stability of the membrane, although the cell volume was increased in the RS047-wl.

The cell wall presents a thick, tough, and elastic structure, which protects the bacteria from the outside environment. In this study, an increase in the expression of several genes and metabolites related to cell wall metabolism was observed, which resulted in an enhanced metabolic activity of the cell wall. As a result, the thickness of the cell wall would not become significantly thinner after the significant increase in cell volume, which improved the heat resistance of bacteria. Lipoteichoic acid (LTA) and peptidoglycan form the cell wall of Gram-positive bacteria [19]. In Gram-positive bacteria, precise control of peptidoglycan synthesis is essential for maintaining cell shape and integrity resisting stresses. YlaN has been reported to be essential for cell survival [20] and is thought to be involved in controlling cell shape [21]. Lacking YlaN resulted in non-uniform cell width [22]. The dipeptidase Pepv was one of the proteolytic enzymes, a host of intracellular peptidases, functioning in the last stages of proteolysis. The dipeptidase PepV acts as a linker between nitrogen metabolism and cell wall synthesis in *Lactococcus lactis*. Deletion of the dipeptidase gene *PepV* in *Lactococcus lactis* resulted in a prolonged lag phase and serious defects in their shape [23].

LTA is a polymer of riboflavin and glycerol residues interconnected by phosphate [24]. It is therefore possible that cell wall synthesis is improved when the glycerol and PA contents increased. In addition, the study showed that LTA adhered to heat-resistant proteins on bacterial cell walls, suggesting that LTA could stabilize heat-resistant proteins in a manner conductive to maintaining heat resistance [25]. The *mraZ* and *mraW* are the head genes of the cell wall gene cluster, which are highly conserved in bacteria (Figure 10). MurF catalyzes the final cytoplasmic step in the synthesis of the bacterial cell wall and is essential for bacterial survival. The cell synthesizes two types of LTAs: wall teichoic acid (WTA) and membrane teichoic acid (MTA). WTAs are exported by the TagGH transporter, which contains TagG and TagH subunits for transport across membranes and into ATPases. During WTA activation, TagG and TagH transport substances across cytoplasmic membranes [26] and provide energy [27]. The lipoteichothenase (LtaS) protein is an integral membrane protein with 5-N transmembrane helices, required to stimulate LTA synthesis, and contributes to bacterial growth [28]. As a folding factor for secreted proteins, PrsA is a ubiquitous 30 kDa lipoprotein found between the plasma membrane and cell wall [29]. When *PrsA* is lost, peptidoglycan crosslinking is negatively affected [30], cell wall integrity is decreased, osmotic shock is more likely to occur, and antibiotic resistance is increased [31]. MltG, a novel lyase found widely in bacteria, has a terminating enzyme activity capable of cleaving nascent chains of glycan to complete the protein–peptidoglycan binding process. It was reported MltG deficient mutants have longer glycan chains than wild-type cells [32]. The protein LysM and LT are two parts of MltG. Most bacterial LysM proteins can bind to various types of peptidoglycan and chitin, recognizing the N-acetylglucosamine, which will impact intercellular action, such as biofilm formation [33]. LysM in certain plant kinases enables the plant to recognize the symbiotic bacteria or sense and induce resistance against fungi. When LysMs binds less efficiently to cells, it results in reduced cell lysis. Then, all LysMs are removed and peptidoglycan binding is lost, concomitant with an almost total loss of peptidoglycan hydrolyzing activity and defense response [34].

In RS047-wl, the activity of lactate dehydrogenase decreased, as well as the metabolites of the tricarboxylic acid cycle and purine metabolism. This resulted in a slower growth rate during the logarithmic stage compared to WT RS047, indicating a delayed growth cycle in RS047-wl. Continuous culture to heat shock led to transient suppression of the prokaryotic translation machinery of the *Clostridium botulinum*, indicating a temporal growth arrest of the strain [35]. Kanshin et al. suggested heat-shock reduced *Saccharomyces* cell cycle progression and represents an adaptive response of yeast cells to environmental stress [36]. Hence, the extended growth cycle of the RS047-wl was also an adaptive response to high temperature.

## 4. Materials and Methods

### 4.1. Changes of Physiological and Biochemical of RS047-wl

#### 4.1.1. Evaluation of Heat Resistance of Strains

Using the same protective agent and carrier coated with bacteria mud, pilot-scaled fermentation of 1-ton fermentation tank (*n* = 3) was conducted for RS047-wl and WT RS047. The survival rate of the obtained product was measured as follows: we weighed 25 g of the product, 17% (4.25 mL) of water. After stirring well, we sealed the triangle bottles with a rubber plug and placed them in a water bath at 85 °C for 10 min. Then, we diluted the heated samples with 220.75 mL of water. We determined the ratio between the heated and unheated products to calculate the survival rate.

#### 4.1.2. Growth Performance of RS047-wl

A single colony from each strain was inoculated into 100 mL of MRS medium and incubated at 37 °C for 24 h. The optical density at 600 nm (OD_600_) and total bacteria count were measured at 2-h intervals (*n* = 3).

#### 4.1.3. Determination of Key Enzyme Activities in Metabolism

*E. faecium* cells were cultured at 37 °C for 12 h and then harvested by centrifugation (3500× *g*, 10 min). They were washed twice with phosphate-buffered saline (PBS) and adjusted to pH 7.2. Then, 1 mL of bacterial solution was taken and broken with an ultrasonic cell crusher (Ningbo Scientz Biotechnology Co., Ltd., Ningbo, China) with an ice bath, power 200 W, ultrasonic 3 s, interval 10 s, repeating 30 times. The cells were treated with the micro enzyme activity kit (Beijing Solarbio Science & Technology Co., Ltd., Beijing, China), then detected with the Enzyme standard instrument (PerkinElmer, Waltham, MA, USA) at the wavelength of 450 nm. The indicators tested include lactic dehydrogenase, hexokinase, pyruvate kinase, Na^+^K^+^-ATPase, and Ca^++^Mg^++^ATPase.

#### 4.1.4. Membrane Fatty Acid Composition Analysis

Fatty acid analysis was performed according to the method of Garces and Mancha [37]. One single colony of WT RS047 and RS047-wl was transferred into test tubes with 1 L of MRS and incubated at 37 °C for 12 h. Then, cells were harvested by centrifugation, washed twice with 0.1 mol/L PBS, and freeze-dried. Accurately, 100 mg drying samples was weighed and resuspended in 4 mL chloroform–methanol (1:1 *v*/*v*), and 2 mL 0.88% NaCl was added and vortexed for 30 s. The solution was centrifuged, and the lower phase was removed and dried under a stream of nitrogen. Then, 4 mL of isooctane and 200 μL Zn-KOH were added as methyl esterification reagent, then 1 g NaHSO_4_ was added. The resulting solution was detected with Agilent J049-2 gas chromatography (Agilent Technologies, Inc., Wilmington, DE, USA). GC settings injector temperature 260 °C, split ratio 100:1, detector temperature 260 °C, air flow in detector 450 mL/min, hydrogen flow 40 mL/min. For each fatty acid, the relative peak area was calculated using the formula: (peak area of one fatty acid/total peak area) × 100%.

#### 4.1.5. Morphological Analysis of RS047-wl

The samples were analyzed using a transmission electron microscope (TEM) (H7500; Hitachi, Tokyo, Japan) to assess changes in the morphology of RS047-wl under heat treatment. After being cultured at 37 °C for 24 h, the cells were washed twice and resuspended in 0.1 mol/L PBS (pH 7.2). The cells were fixed with 1% osmium for 24 h then dehydrated in ascending concentrations of ethanol and air-dried at room temperature. After embedding, the samples were sectioned into ultra-thin sections.

### 4.2. Proteomic Analysis

#### 4.2.1. Total Protein Extraction

All experiments were performed in triplicate. A total of 250 mL culture was collected after being cultured for 12 h and transferred to a pre-weighed 50 mL conical tube, then centrifuged at 7500 rpm for 10 min at 4 °C. Then, the samples were stored at −80 °C and sent to Shanghai Majorbio Bio-pharm Technology Co., Ltd. (Shanghai, China) adding an appropriate amount of buffer (200 mM DTT, 1% SDS), which contains protease inhibitors to inhibit protease activity. Then, we carried out an ultrasound for 2 min and splitting for 30 min. After centrifugation at 12,000× *g* under 4 °C for 30 min, the supernatant was determined by the bicinchoninic acid (BCA) method by BCA Protein Assay Kit (Thermo Fisher Scientific, Pierce, CO, USA). Equal amounts of protein were loaded into each lane on an SDS-PAGE gel for electrophoresis analysis.

#### 4.2.2. Digested and Tagged of Proteins with TMT Reagent

We added TEAB (triethylammonium bicarbonate buffer) to 100 μg protein samples to 100 mM. Then, we added TCEP (tris (2-carboxyethyl) phosphine) to 10 mM and let the reaction take place for 40 min under dark conditions. Then, 6 times the acetone was added to the sample and settled for 4 h at −20 °C. The sediment was collected after centrifugation for 20 min at 10,000× *g* and 100 µL 100 mM TEAB was added. Finally, samples were digested with a 1:50 trypsin-to-protein mass ratio overnight at 37 °C. One unit of TMT reagent was added to a 100 μg sample for 2 h at room temperature. Then, hydroxylamine was added to react for 30 min at room temperature. The concentrated samples were fractionated into fractions by Vanquish Flex UPLC (Thermo Fisher Scientific, Pierce, CO, USA) with ACQUITY UPLC BEH C18 Column (1.7 µm, 2.1 mm × 150 mm, Waters, Milford, USA) to increase proteomic depth.

#### 4.2.3. Mass Spectrometry

Online liquid chromatography–tandem mass spectrometry was performed on an Evosep One system (Evosep, Odense, Denmark) connected to an Orbitrap Exploris 480 (Thermo Fisher Scientific, Pierce, CO, USA) with a nanoelectrospray ion source. The samples were analyzed using nanoflow liquid chromatography–tandem mass spectrometry on an Evosep One system (Evosep, Odense, Denmark) connected to an Orbitrap Exploris 480 (Thermo Fisher Scientific, Pierce, CO, USA) via a nanoelectrospray ion source. MS spectra were analyzed with the Orbitrap with 60,000 resolutions. The MS/MS resolution was set to 15,000 (at fixed first mass *m*/*z* 110), the smallest automatic gain control target at 8 × 10³, and the maximum fill time at 22 ms.

#### 4.2.4. Database Searching and Data Analysis

In ProteomeDiscoverer (Thermo Scientific, Version 2.4), the RAW data files were analyzed against the CDs.faa_unique.fasta database with a false discovery rate (FDR) of 0.01 for peptide identification. The differentially expressed proteins were identified using fold changes (>1.2 or <0.83) and *p*-values < 0.05. GO (http://geneontology.org/, accessed on 3 September 2022) and KEGG (http://www.genome.jp/kegg/, accessed on 3 September 2022) pathways were used for pathway annotations. In addition, DEPs were used for GO and KEGG enrichment analysis. Protein–protein interaction analysis was performed using String v10.5. Data were analyzed through the online platform of Majorbio Group (cloud.majorbio.com, accessed on 10 September 2022).

### 4.3. Detection of Transcriptomic and Metabolomic

The detection methods of transcriptome and metabolome were mentioned in Wang [15].

### 4.4. Statistical Analyses

The statistical analysis for the survival rate, metabolic enzymes, cell characteristics, and fatty acid contents were performed with SPSS 22.0 software (SPSS Inc., Chicago, IL, USA). Data are presented as mean ± SEM), and the one-sample Kolmogorov–Smirnov test was used for the normality test. Differences in continuous variables of the survival rate, metabolism enzymes, cell characteristics, and fatty acids contents were assessed by the *t*-test. Statistical tests were two-tailed and *p* values < 0.05 were considered statistically significant.

## 5. Conclusions

We conducted experimental verification and found that the resistance of RS047-wl to heat stress was significantly improved. Through physiological and multi-omics analyses, we observed that the RS047-wl strain experienced an increase in intracellular molecular distances, leading to a subsequent significant increase in cell volume when exposed to high temperatures. In order to maintain its cell wall and membrane thickness as well as normal morphology, RS047-wl enhanced its capabilities for cell wall and membrane synthesis, thereby improving its ability to withstand high temperatures. Furthermore, the increase in proportions of saturated fatty acid within the cell membrane contributed to strengthening intermolecular interactions, thereby improving the strain’s resilience against high temperatures.

## Figures and Tables

**Figure 1 ijms-24-11822-f001:**
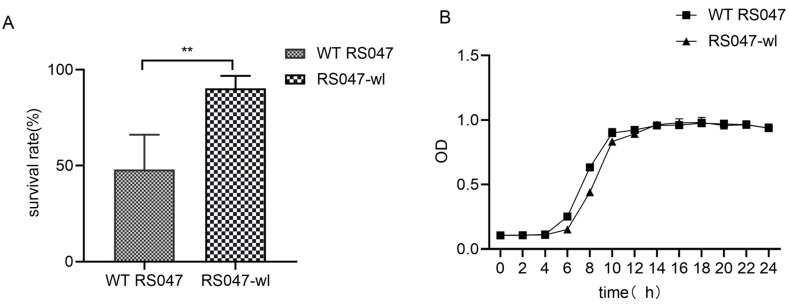
The growth performance and the resistance against high temperature of WT RS047 and RS047-wl. (**A**) Survival of WT RS047 and RS047-wl exposed to 85 °C water bath for 10 min. (**B**) Growth curves of WT RS047 and RS047-wl in MRS medium, at a temperature of 37 °C. Data are represented as mean ± SEM (*n* = 3). (** *p* < 0.01).

**Figure 2 ijms-24-11822-f002:**
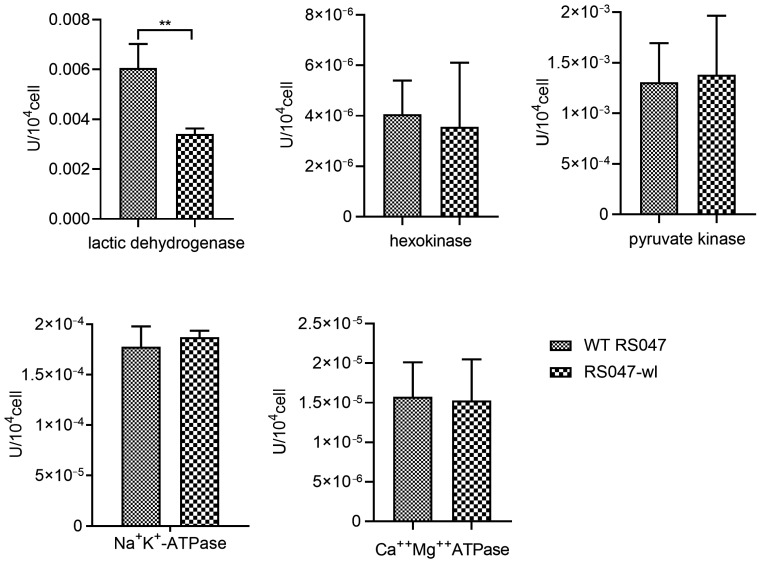
Key metabolic enzyme activities in WT RS047 and RS047-wl. Data are represented as mean ± SEM (*n* = 3). (** *p* < 0.01).

**Figure 3 ijms-24-11822-f003:**
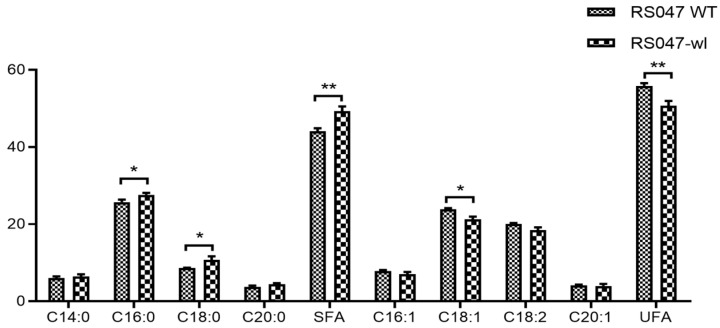
Main fatty acids of the WT RS047 and RS047-wl. Data are represented as mean ± SEM (*n* = 3). (* *p* < 0.05; ** *p* < 0.01).

**Figure 4 ijms-24-11822-f004:**
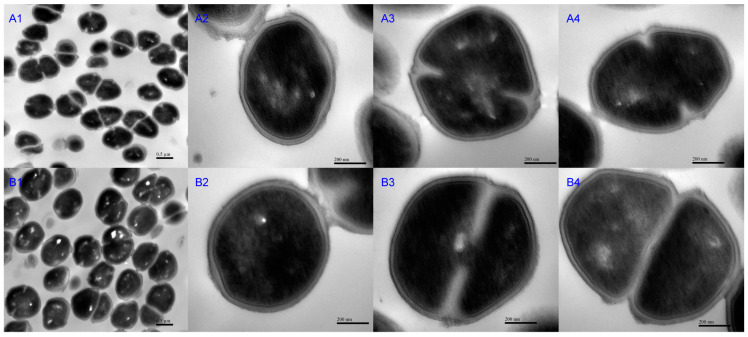
Cell morphology under transmission electron microscope. A: WT RS047, B: RS047-wl. The cells were divided into three stages: undivided period (**A2**,**B2**), early fission period (**A3**,**B3**), and metaphase and anaphase period (**A4**,**B4**). The magnification for (**A1**,**B1**) was 25,000×, for (**A2**–**A4**,**B2**–**B4**) was 120,000×.

**Figure 5 ijms-24-11822-f005:**
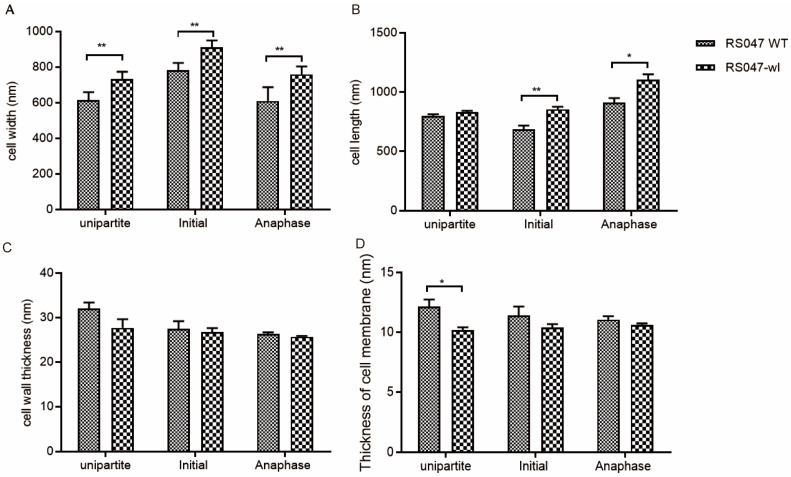
Cell morphology characteristics of WT RS047 and RS047-wl. (**A**) Cell width. (**B**) Cell length. (**C**) Cell wall thickness. (**D**) Thickness of cell membrane. Data are represented as mean ± SEM (*n* = 3). (* *p* < 0.05; ** *p* < 0.01).

**Figure 6 ijms-24-11822-f006:**
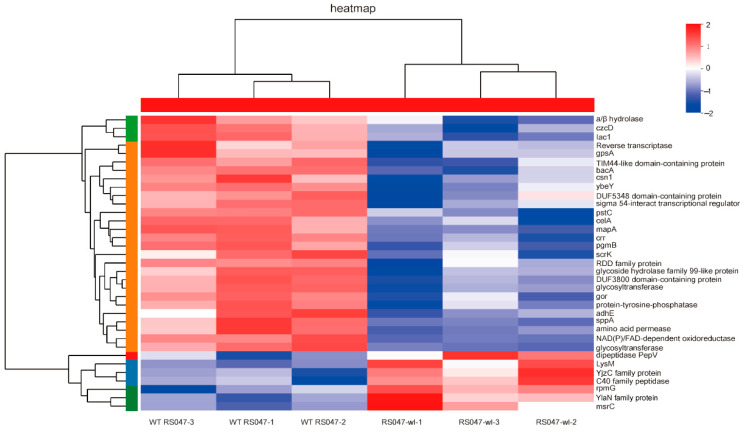
Heat map analysis of significantly changed proteins. The colors of the column indicate the enrichment significance, and the red represents significantly upregulated proteins and blue represents significantly downregulated proteins.

**Figure 7 ijms-24-11822-f007:**
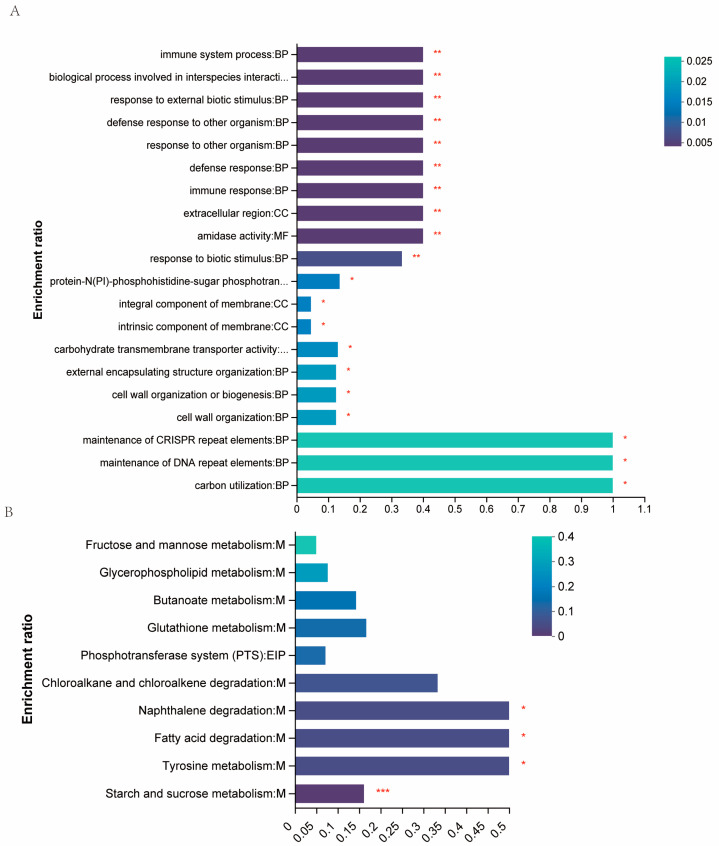
The statistical enrichment of differential expression proteins. (**A**) The statistical enrichment of differential expression proteins in Go pathways. The y-axis shows the GO term. The x-axis shows concentration enrichment ratio, referring to the ratio between the protein number enriched in the GO term and the background number annotated to the GO term. The colors of the column indicate the *p*-value. (**B**) The statistical enrichment of differential expression proteins in KEGG pathways. The y-axis shows the pathway name. The x-axis shows the concentration enrichment ratio, referring to the ratio between the protein number enriched in the pathway and the background number annotated to the pathway. The colors of the column indicate the enrichment significance, the darker the default color, the more significant the enrichment of the KEGG term. (* *p* < 0.05; ** *p* < 0.01; *** *p* < 0.001).

**Figure 8 ijms-24-11822-f008:**
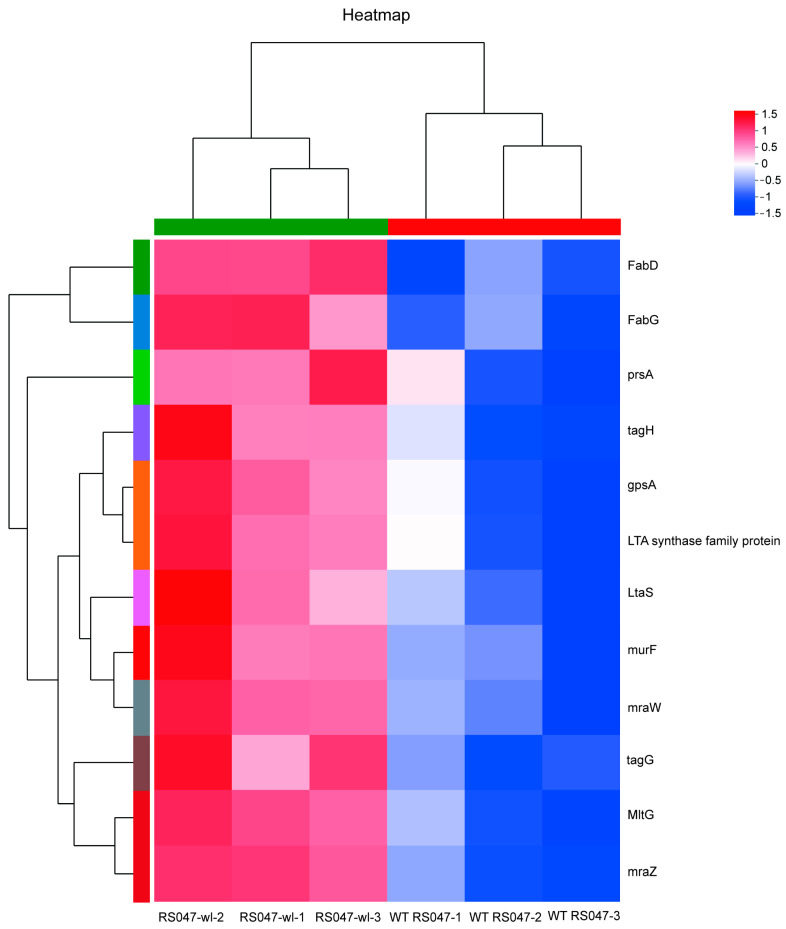
Heat map analysis of significantly changed genes related to cell wall and cell membrane metabolism. The colors of the column indicate the enrichment significance, and the red represents significantly upregulated genes and blue represents significantly downregulated genes.

**Figure 9 ijms-24-11822-f009:**
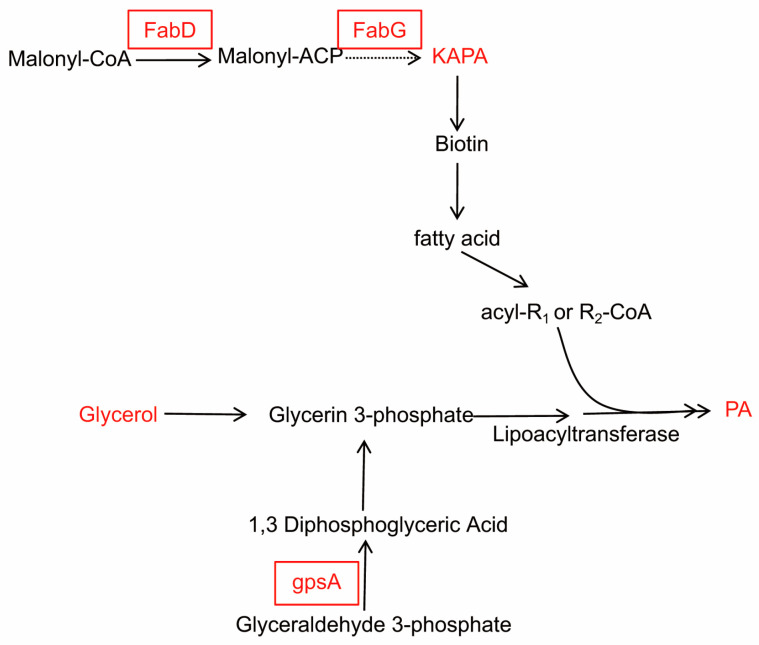
Cell membrane metabolism changes. The red letters represent significantly upregulated genes (with band box) and metabolites (no band box) in RS047-wl compared to WT RS047. The black letters represent no significant change between RS047-wl and WT RS047. KAPA: 8-Amino-7-oxononanoate; PA: phosphatidic acid.

**Figure 10 ijms-24-11822-f010:**
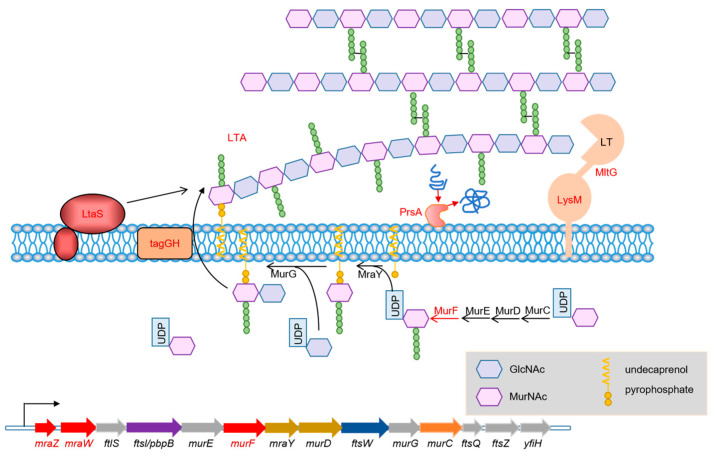
Cell wall metabolism changes. The red letters represent significantly upregulated genes and proteins in RS047-wl compared to WT RS047. The black letters represent no significant change in RS047-wl and WT RS047. LTA: lipoteichoic acid; UDP: uridine diphosphate; GlcNAc: N-acetylglucosamine; MurNAC: N-acetylmuramic acid.

## Data Availability

The data presented in this study are available upon request from the corresponding author.

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
