# Peer review of "Enhanced Cell Wall and Cell Membrane Activity Promotes Heat Adaptation of Enterococcus faecium"

_ijms, 2023, doi:10.3390/ijms241411822_

Round 1

Reviewer 1 Report

This is a very significant study. The study is well design and the results support the conclusion. I have two comments that should be addressed.

In the first sentence of the abstract the authors say "Enterococcus faecium (E. faecium) are suggested as satisfactory antibiotic alternatives to animals by improving growth performance, nutrient digestibility and preventing or treating diarrhea.", however the sentence must be clarified in order to better understanding the aim of the work.

Figure 5: It was not possible to see Figure 5. It would be good to be able to see it.

This is a very elegant study that after this small changes should be published.

Reviewer 2 Report

I have carefully read the article and I have noted in the pdf text that I am attaching some writing errors or omissions. I also commented on an inversion of two chapters (Materials and methods and Results) which I reproduce in the pdf text. I ask the Authors to take note of my observations.

The article has only a few small errors to correct, but it is readable even if very technical as in English.
